# Residents' Selection Behavior of Compensation Schemes for Construction Land Reduction: Empirical Evidence from Questionnaires in Shanghai, China

**Jianglin Lu [1], Keqiang Wang [1,2] and Hongmei Liu [3,***

1   School of Public Economics and Administration, Shanghai University of Finance and Economics, Shanghai 200433, China
2   Technology Innovation Center for Land Spatial Eco-Restoration in the Metropolitan Area, MNR, Shanghai 200003, China
3   School of Finance and Business, Shanghai Normal University, Shanghai 200233, China
*   Correspondence: hmliu@shnu.edu.cn

**Abstract:** Construction land reduction (CLR) was implemented in China to improve the efficiency of construction land use. CLR also limited the development of net reduction areas of CLR. By analyzing the Task-Quota-Financial-Benefit flow of CLR, this paper proposes three typical compensation schemes and uses the multivariate probit model to study residents' selection behavior for these schemes. It is found that (1) in order to compensate for the losses caused by CLR to the reduced direct subjects, there can be three types of possible schemes: direct economic compensation (Scheme I), in situ (Scheme II) and off-site (Scheme III) enhancement of development capacity. (2) The more reasonable the compensation standard, the greater the employment pressure in the township and the greater the township's location disadvantage, which is why more residents prefer Scheme III. (3) The higher their family income and their family support pressure, the more they prefer Scheme III. (4) At this stage, there is no significant difference in the choice of compensation schemes between cadres and non-cadres. (5) The net planning reduction area prefers Scheme I, while other areas prefer Scheme II. The conclusions may provide insight into the demand for more reasonable compensation policies to ensure the sustainability of CLR.

**Keywords:** construction land reduction; compensation; residents' selection behavior

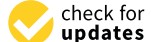



## 1. Introduction

The suburban areas in economically developed regions of China are still in a stage of rapid urbanization and development, and the contradiction between the supply and demand of construction land remains very prominent [1–3]. China has implemented a system of spatial use planning and land use control and controls the total amount and intensity of construction land. In 2016, the *Outline of the 13th Five-Year Plan for Land Resources* emphasized the need to revitalize the stock of construction land, "implement the control of the total amount of construction land and the management of reduction" [1], which has effectively controlled the disorderly expansion of new cities and new development zones expansion. However, the control of construction land constrains further economic development, especially in the suburban areas of economically developed regions. The contradiction between the supply and demand of construction land is very prominent [1,2,4].

The contradiction between the supply and demand of construction land can be achieved by improving the quality and efficiency of the stock of construction land [5], or by implementing the reduction of inefficient construction land [6]. The latter, namely construction land reduction (CLR), is a land restoration tool that reclaims inefficient, dispersed and heavily polluted construction land outside urban concentrated construction areas and transforms such land into cultivated land or ecological land and generates an

equal amount of land quota for construction purposes [2,6–9]. These construction land quotas can be used in new areas with higher efficiency [7,8]. Generally speaking, these construction land quotas will be used for centralized construction areas. According to China's control over the total amount and intensity of construction land, the increase in construction land area is extremely limited. For every unit of cultivated land occupied by construction land, one unit of cultivated land must be replaced. CLR has converted construction land into cultivated land or ecological land. Therefore, when using the construction land quota obtained through CLR, there is no need to supplement the amount of cultivated land. That is to say, CLR produces a balanced quota of cultivated land occupation and compensation, and it also creates the development space for new construction land. CLR thus provides a source of quota for new construction land through the transfer of existing construction land [2]. This is a method to obtain construction land quota under the control of the total amount of construction land [7,8]. In urban areas of economically developed regions, it is mainly the former [2,8,10], while in suburban areas, it is mainly the latter [6,7]. In economically developed areas, the demand for construction land is not enough to be met by improving the quality and efficiency of the stock of construction land, so it is particularly important to free up space for construction land through CLR. In economically developed regions, there is a new trend in land use patterns from an incremental expansion of construction land to a reduction of the stock of construction land [11]. As early as 2013, Shanghai actively began to explore CLR, and it was the first province in China to implement CLR policies on a region-wide scale [2]. Beijing has also experimented with CLR, and according to the *Urban Master Plan of Beijing, China (2016–2035)* [2], Beijing regards townships as the basic unit. Beijing has established a push-back mechanism by linking the addition of urban construction land and the vacating of collective construction land proportionally.

While CLR solves the contradiction between the supply and demand of construction land, it also causes a slowdown in the development of net reduction areas of construction land (NRACL) and a change in the interest demands of residents in CLR areas [9,11,12]. After the industrial land reduction, the town-level tax revenue and the rental income of the village collective economic organizations will be reduced [2], seriously influencing regional economic development [9]. This dilemma stems from the spatial allocation of land quotas obtained through CLR. CLR provides space for new construction land through the reduction and reclamation of current inefficient construction land into arable land, generating new construction land quotas and arable land occupation balance quotas. At the present stage, the allocation of CLR and the land quotas formed by CLR is based on the principle of efficiency, i.e., in the process of reduction, areas with poor locations are given priority to reduce and become NRACL, while in the process of the allocation of land quotas formed by CLR, areas with superior locations are given priority to obtain construction land quotas and become net increase areas of construction land (NIACL). NIACL is the area with an advantageous location. Through the reduction of inefficient construction land outside the centralized construction region and using the reduced quota for land development in the centralized construction region with comparative advantages, the rapid development of the centralized construction region will be realized [8]. The core of CLR is the "reduction" of inefficient construction land outside the centralized construction region and the use of the reduced quotas for the development of land within the centralized construction region [13]. In this process, NRACL has lost some development opportunities, i.e., it is difficult to give full play to the advantage of backwardness and secure development benefits in the allocation process of land quotas formed by CLR. CLR seriously influenced the livelihood of local residents [9], and this has affected the support and implementation of CLR policies by the residents in NRACL. In order to meet the needs of urbanization and high-quality development, it is necessary to continue to promote the implementation of the CLR policy, then it is necessary to pay sufficient attention to the realization of the development interests of NRACL. In order to solve the development dilemma of NRACL, compensation for NRACL is needed. CLR involves the interests of both macro-level

reduction subjects (municipal government, district governments, township governments, and village collectives) and micro-level individuals (land enterprises and residents). Since the implementation of the CLR policy in Shanghai in 2014, some changes have occurred in the compensation for the loss of interests of CLR-reduced direct subjects. In the early stage of the implementation of the CLR policy, the direct loss of interests was used as the basis of compensation, and the objects of compensation were the directly damaged subjects, without considering the indirect losses and indirectly damaged subjects; the basis of being compensated and the objects of compensation were not comprehensive. With the expansion of the scale of the CLR and the extension of the period, its indirect impacts are becoming more and more prominent, especially the loss of local employment opportunities [2,9,14]. The decline of sustainable development potential leads to the change of demands of the residents in NRACL. Residents in NRACL propose that the NRACL reserve more construction land quotas to improve the development potential of the NRACL. They also propose that the transfer of the quotas obtained from the CLR in the NRACL should be subject to the NRACL's population migration. In order to further compensate for the loss of the interests of the reduced areas, a land value-added income-sharing mechanism should be established [15]. Due to the lack of experience in performing CLR, enterprises and residents that have been affected did not obtain sufficient compensation [9]. The demands of NRACL's residents affect the cost and form of compensation, which in turn affects the costs and benefits of the NIACL's use of CLR quotas and implicates changes in their attitudes toward CLR. Thus, compensation schemes are something that affects both NRACL and NIACL. Different compensation schemes have different impacts on residents in different regions and scenarios.

The compensation for rural homestead withdrawal is crucial to the reconstruction and sustainable development of farmers' livelihoods [16]; however, few scholars have studied the compensation behavior of CLR. This paper explores the selection behavior of residents in CLR areas for compensation schemes for NRACL by taking residents in the area where CLR is implemented as the research object. This paper firstly conducts a theoretical analysis of residents' selection preferences for different compensation schemes in CLR; secondly, it uses the multivariate probit (MVP) model to test residents' selection preferences for compensation schemes empirically, and, lastly, it draws policy implications for optimizing CLR compensation.

The rest of the paper is organized as follows: the second part is a literature review; the third part is a theoretical analysis and research hypothesis; the fourth part is the research methodology and data; the fifth part is the empirical results and analysis; the last part comprises the conclusion and policy implications.

## 2. Literature Review

According to John Friedmann, urbanization is a dynamic, multidimensional socio-spatial process [17]. Urbanization is also a process of construction land expansion that involves the concentration of population to cities [18–20]. In Lewis' dualistic economic structure theory [21], it is assumed that urban and non-agricultural land is satisfied and unconstrained during the transfer of surplus rural labor to cities and non-agricultural industries. However, in a country with a large population like China, strict control of construction land has been implemented for the sake of arable land protection and ecological civilization, and it has become increasingly difficult to rely on construction land expansion to meet economic development needs [1,2,4]. The reduced development model under the control of the total amount and intensity of construction land has been promoted, and CLR has become an important means to control the uncontrolled expansion of construction land and achieve sustainable development [2,8,9], which has begun to receive attention from scholars.

Pre-existing studies have mainly focused on the economic impact of CLR, mechanisms and stakeholder interests.

(i)　Studies on the economic impact of CLR. CLR is a path innovation to break the constraint of tight construction land quotas [22] and meet the demand for construction land for economic and social development through the optimization of construction land use structure [6–8]. Some studies suggest that the positive effect of industrial land reduction on local fiscal revenue growth gradually increases over time [22]. One study analyzed the impact of homestead reduction on economic agglomeration and rural revitalization and found that homestead reduction can significantly increase the income of rural residents and promote industrial integration and rural economic development; moreover, this positive impact intensifies over time [23].

(ii)　Studies on the mechanisms of CLR. Some studies analyzed the impact of CLR on rural transformational development based on the grounded theory and concluded that resource allocation is the core impact of CLR on rural transformational development [15]. Some studies have also focused on the land marketization mechanism [24] and operation mechanism [25] in the CLR process, etc.

(iii)　Studies on the benefits of CLR to stakeholders. Some studies have focused on the impact of industrial land reduction policies on town-level and village-level interests [2], location selection [8], the mechanism and policy of industrial land lifecycle management [26], etc.

(iv)　Research on land consolidation issues. Land consolidation is the most complex, technical and important stage of land reclamation [27]. Land consolidation is a spatial planning process [28,29], and this process of reorganizing property rights is often the main cause of dissatisfaction and opposition to land remediation [30]. Land consolidation inevitably leads to changes in land ownership and adversely affects the interests of landowners steadily, thus leading to controversy and discontent [27]. Some studies focused on the barriers encountered in CLR to meeting the sustainability requirements of land use [31].

In summary, the established literature has mainly studied the economic impacts, mechanisms and stakeholder interests of CLR. The fundamental goal of development is to continue to improve people's well-being [32]. The purpose of CLR is to develop and improve people's well-being; however, the current operation of CLR has brought about the problem of unfair distribution of benefits [9], especially unequal opportunities for development and damage to the interests of residents in NRACL. These problems have affected the implementation of the CLR policy, and therefore the compensation policy needs further improvement. Policymakers may be faced with the following questions: What are the expectations of residents regarding compensation for CLR? What are the available compensation schemes? Residents' choices of these compensation schemes may be heterogeneous in terms of regional conditions, individual resident characteristics, household characteristics, etc. How can compensation schemes be matched to these heterogeneities? There is a gap in the existing literature when it comes to CLR and even less on compensation for the relevant stakeholders in the CLR process. For this reason, this paper focuses on the behavior of residents' choices of compensation schemes in the CLR process. Through the research questionnaire, we analyze the residents' preference for the choice of compensation scheme in the CLR process and propose insights for improving the compensation policy of CLR. Since CLR is now carried out mainly for collective construction land, the reduction of state-owned construction land, especially inefficient construction land of state-owned enterprises outside the planning area, is more resistant and less carried out. Thus, the object of this paper is the CLR of collective construction land. Meanwhile, Shanghai, as a region that has taken the lead in conducting region-wide CLR since 2014, has a relatively well-developed CLR policy process. Therefore, this paper mainly uses Shanghai as an example and conducts a theoretical analysis of the various levels of government and underlying subjects involved in the policy operation.

The possible marginal contributions of this paper are mainly the following: first, by studying the task flow, quota flow, financial flow and benefit flow in the CLR process, the research hypothesis of benefit realization schemes and influencing factors of the direct

subjects being reduced in the CLR process is proposed. Second, based on micro-research data from several areas in Shanghai, an MVP model that can handle multiple binary choices is simultaneously constructed and empirically tested, and the heterogeneous effects of resident status and the types of land use planning are identified. Third, based on the findings of the study, policy implications for improving the compensation scheme of CLR are proposed.

## 3. Theoretical Analysis and Research Hypothesis

### 3.1. Task, Quota, Financial and Benefit Flows in the CLR Process

The CLR process involves three levels of government—municipal, district and township—and three types of underlying subjects [11,14]. Village collectives, land enterprises and residents are the base subjects most affected by CLR [9,15]. In the CLR process, the tasks of CLR are decomposed from top to bottom (municipal government → district governments → township governments → underlying subjects), and the land quotas formed by CLR are collected, managed and deployed for use from bottom to top (underlying subjects → township governments → district governments → municipal government), while the compensation funds for CLR are passed and accumulated from top to bottom (municipal government → district governments → township governments → underlying subjects) for passing and accumulating at each level. See Figure 1 for details.

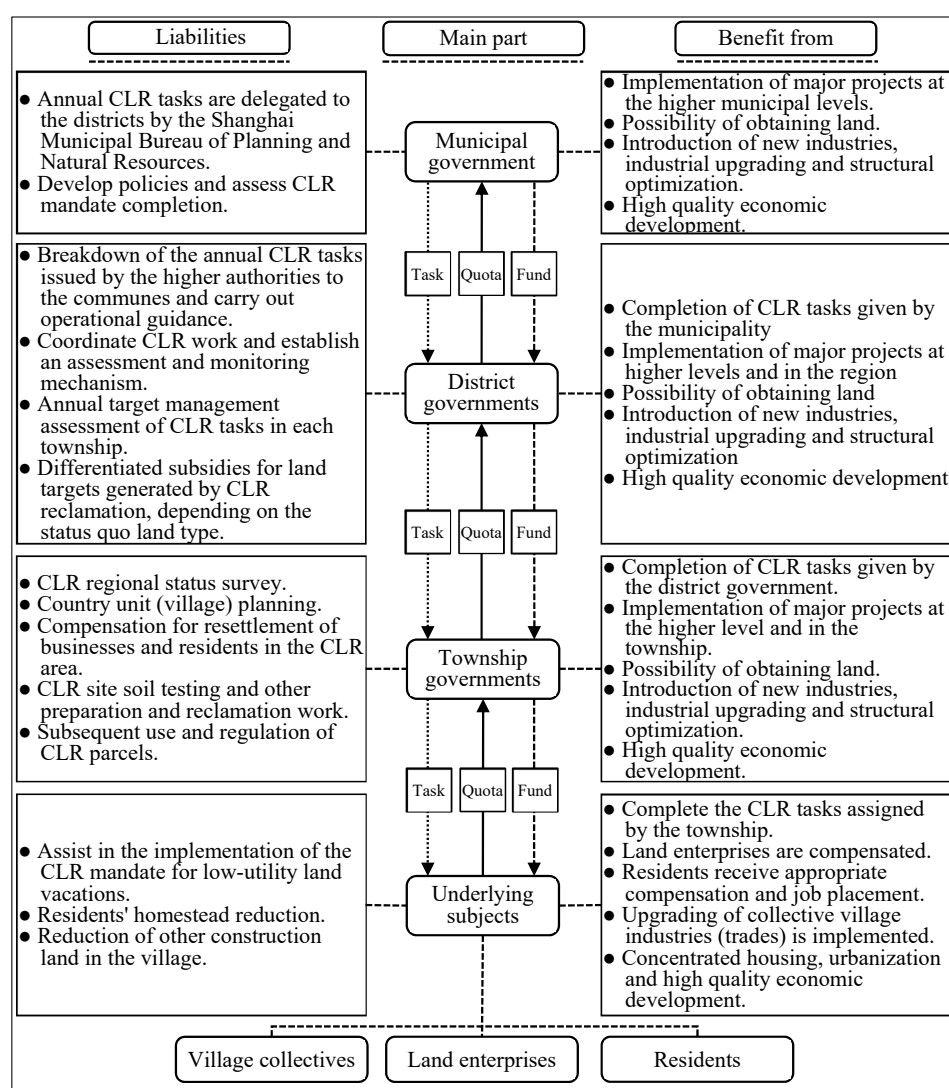

**Figure 1.** Task, quota, financial and benefit flows in the CLR process.

### 3.1.1. Task Flow and Quota Flow in the CLR Process

Shanghai Municipal Bureau of Planning and Natural Resources will delegate the annual CLR tasks to each district based on the medium and long-term plans, near-term plans, three-year rolling plans and annual plans of each district. Based on the annual CLR tasks issued by the Shanghai Municipal Bureau of Planning and Natural Resources, the district governments decompose and distribute the annual CLR tasks to the townships based on the medium- and long-term plans, near-term plans, three-year rolling plans, and annual plans of the townships and carry out operational guidance. The municipal governments formulate policies and assess the completion of CLR tasks. The district governments coordinate CLR work, establish an assessment and supervision mechanism, conduct annual target management assessments on the completion of CLR tasks in each township and implement differentiated subsidies for land quotas generated by CLR reclamation according to the status quo land type. Township governments conduct surveys on the current status of CLR, carry out country unit (village) planning, compensate for the resettlement of relevant enterprises and residents and carry out soil testing and other finishing and reclamation work. The townships implement finishing and reclamation plots according to the spatial planning of the countryside, land finishing planning and plans for new arable land for land development and reclamation and carry out the subsequent use and supervision of land quotas obtained by CLR. Basic subjects assist in implementing CLR tasks according to township plans, including the reduction of the residents' homesteads, the reduction of other construction land in villages and the vacating of low-utility land.

According to the relevant provisions of the *Land Management Law* of the People's Republic of China, all types of construction land projects that occupy agricultural land and unused land are subject to land allocation targets. Thus, in areas where construction land is close to the "ceiling", the allocation of construction land is "reduced to increase" under planned land use control. Reduction to increase means that under the control of the total amount and intensity of construction land, the new construction land quota is subject to CLR, and without CLR, no new construction land can be added. State-level projects and major urban projects are not subject to this restriction. The construction land quotas created through the CLR include the new construction land turnover quota and the arable land occupation balance quota (referred to as "dual quotas"). The new construction land turnover quota (referred to as the "single quota") is obtained through the CLR after the land is reclaimed as agricultural land or unused land; the arable land occupation balance quota is obtained through general reclamation and CLR after the land is reclaimed as arable land. According to the policy document and practice, each township government, after completing the CLR tasks assigned by its district, can include land quotas not exceeding about 25% of the annual CLR acceptance volume into the township quota account, which can be used to meet the demand for new construction land quotas for each township's own construction projects. Moreover, this portion of land quotas will no longer be collected and stored by the district government, no municipal and district funding subsidies will be made and no quota use fees will be charged. The "dual quotas" obtained from the CLR plots of each township after confirmation of acceptance are subsidized in accordance with the standards and are, in principle, collected and stored by the district government, managed and deployed for use.

### 3.1.2. Financial Flows for Compensation in the CLR Process

In order to motivate the lower level governments to CLR, the higher level governments will subsidize the lower level governments by a certain amount, including direct subsidies, incentives for intensive land use, incentives for timeliness, incentives for over-completion of tasks and subsidies for relocation. Township governments receive incremental land proceeds and return benefits to the relevant subjects on CLR land. Municipal and district governments use part of the land grant revenue to compensate CLR projects. The compensation for industrial land is for the land enterprises, the compensation for residential land is for residents and the compensation for construction land reduced by other village

collectives is for village collectives. Pre-CLR costs include compensation for relocation and land reclamation fees for the demolition of buildings on the current construction land, costs for farmland facilities and other costs and expenses directly related to the CLR land. The costs for village collectives in the CLR process are mainly the pre-reduction rental income of the enterprise and the transaction costs in the process. Based on the implementation in the Jinshan District, Shanghai, China, it appears that for the licensed parcels the valuation is made to determine the compensation price, while for the unlicensed parcels no compensation is made and the residual value of the buildings on the ground is determined as appropriate.

From the viewpoint of the land quota and source of compensation:

(1) The land quota of the land user or village is collected and stored in the township, and the land user or village collective will receive full compensation. The compensation comes from the township governments, including compensation funds from the municipal, district and township governments. It is used to compensate for the relocation of the site enterprise and other cost expenses of the village collective to carry out CLR.

(2) For township quotas collected in the district, the townships receive compensation and any surplus is used for other CLR work. The compensation funds come from the district government, including compensation funds from both municipal and district governments and are only part of the cost of CLR.

(3) For the district construction land quotas storage to the municipal government, the municipal government gives the district a subsidy of 450,000 CNY/mu (1 mu ≈ 0.067 hectares), and the shortfall is borne by the district and the township themselves. As industrial land is generally offered at low prices, the Government, in order to encourage the use of CLR quotas for the development of the real economy and industries, has increased the subsidy of 150,000 CNY/mu on top of 450,000 CNY/mu for each district to use the land quotas formed by CLR in the "198" area [3] for certified industrial projects. Moreover, the land quotas formed by the CLR in the "198" area are coordinated by the Shanghai Municipal Bureau of Planning and Natural Resources, and about 25% of the land quotas will be used for national and municipal projects. The municipal government does not subsidize the reduction of residential land.

### 3.1.3. Stakeholder Benefit Flows in the CLR Process

Stakeholders in the CLR process include municipal government, district governments, township governments, land enterprises, residents and village collectives. Each stakeholder fulfills its responsibilities and receives corresponding benefits, which can be the acquisition of land quotas or financial compensation. Overall, the underlying subjects, township governments and district governments complete the CLR tasks assigned by the higher-level government and implement major projects at the higher level and major projects in the region, respectively. Municipal government implements major projects at the national and municipal levels; district governments implement major projects at the national, municipal and district levels; and township governments implement major projects at the national, municipal, district and township levels. These projects include industrial projects, infrastructure projects, residential projects, etc. In the CLR process, township and above governments are given the opportunity to add new construction land quotas, which are deployed at the district level, and townships are given the opportunity to partially "reduce to increase". The district can allocate 75% of the CLR quotas, and the degree of "reduce to increase" is stronger. The municipality, as the author of the CLR, takes the initiative and achieves the strongest degree of CLR. Whether the quotas are allocated at the township, district or municipal level, they will be used for the introduction of new industries, industrial upgrading and structural optimization, and high-quality development. For village collectives, in addition to the compensation for the relocation of CLR projects, the surplus funds can be used for the residents' social security, rural development,

township and village infrastructure, the construction of public facilities, and support for the collective development of villages.

The core of CLR is to achieve structural optimization of the construction land space. Structural optimization involves the issue of the radius of optimization. From a macro perspective, it is to expand the radius of the allocation of CLR quotas. (1) Structural optimization is divided into three levels according to the current practice: optimization in the CLR township, optimization in the CLR district and optimization in the city. (2) In terms of the radius of the CLR quota allocation, the larger the radius, the greater the value-added potential of the allocation. Therefore, it can be considered that the potential of CLR quota allocation is the potential for optimization within the CLR's own township $\leq$ the potential for optimization within the CLR's own district $\leq$ and the potential for optimization within the city. Thus, in terms of allocation potential, the radius of the allocation of CLR quotas should be expanded.

As a practical matter, the city, in order to protect the development space of each district, requires in principle that each district uses quotas derived from its own CLR. Therefore, the current use of CLR quotas is freely allocated within the district by district coordination. The proportion of construction land at the discretion of the townships is not high. There have been two rounds of policies, the first two of which did not give townships discretionary CLR targets. In the third round of CLR policies, conducted since 2021, individual districts, such as Jiading, Shanghai, have left approximately 25% of CLR targets at the townships' discretion. In the second round of CLR, 25% of the quotas were achieved across districts, and the specific allocation was coordinated within the municipality, rather than freely traded between districts. This shows that there has been an expansion of the radius of trading in the allocation of CLR quotas. However, this enlargement is configured by the municipal government. In order to protect the development interests of NRACL, certain compensatory measures are required.

### 3.2. Compensation Scheme for the Loss of Development Benefits of NRACL in the CLR Process

Based on the above theoretical analysis and combined with the CLR practice in Shanghai, China, this paper summarizes three compensation schemes that can help promote the realization of the development interests of NRACL in the CLR process.

In daily life, people's choice behavior is often influenced by their preference for alternative options [33]. For this reason, this paper analyzes the typical characteristics of various compensation schemes, as shown in Table 1.

Table 1. Compensation scheme for the benefits of reduction in CLR.

| Scheme Type | Content | Interest Claims | Specific Measures | Benefit Characteristics | Beneficiary Flows |
|---|---|---|---|---|---|
| Scheme I | Direct compensation for NRACL's direct subject losses | Full compensation | (i) Increased compensation rates (ii) Expanding the scope of compensation | (i) Fewer beneficiary subjects (ii) Access for both advantaged and disadvantaged groups (iii) Zero resettlement costs in the areas where the quotas are used, low institutional friction for cooperation and high direct compensation costs (iv) Compensation is one-time and lacks continuity (v) Short and quick | (i) In terms of the allocation of construction land quotas, the NIACL benefits the most in Scheme I; the NIACL benefits the second most in Scheme III; and the NIACL benefits the least in Scheme II (ii) From the point of view of society-wide benefits, society as a whole benefits the most under Scheme II, followed by Scheme I and the least under Scheme III (iii) In terms of the long-term development of NRACL, Scheme II benefits the most; Scheme I the second most; and Scheme III the least (iv) In terms of short-term development of NRACL, Scheme I benefits the most; Scheme II the second most; and Scheme III the least |
| Scheme II | Increasing the scale of development of non-agricultural industries in NRACL, improving the capacity and competitiveness of industries | (i) Increasing employment capacity (ii) Improve sustainable development capacity | (i) Increase the use percentage of land quotas obtained by CLR in NRACL (ii) Assistance for productive capacity enhancement by the regions using the quotas obtained by CLR in NRACL | (i) Wide range of beneficiary subjects (ii) Greater enjoyment by advantaged and disadvantaged groups as well (iii) CLR regions have a strong capacity for autonomous control in the development process (iv) Zero resettlement costs borne by the quotas-using region, and low direct compensation costs in the institutional friction of cooperation (v) Compensation is continuous and has a long cycle time (vi) Local upgrading of development capacity and local urbanization (vii) Low level of technology, management compared with Scheme III, high risk of benefit | |
| Scheme III | Co-development type of off-site development with the transfer of quotas, in which income is increased and employment is expanded in the process of off-site development | Off-site use of construction land quotas with corresponding movement of employment and general population and consequent enjoy the benefits of NIACL | (i) Consequent migration of employed persons (according to certain requirements) (ii) As the employed person moves, the household and social security of his or her family members are also enjoyed | (i) High probability of enjoyment by advantaged groups, with quotas using regional demand for competitive human resources (ii) Increased and larger resettlement costs in the areas where the quotas are used (iii) High institutional frictions in NRACL and NIACL cooperation and low direct one-time compensation costs (iv) Long-term security of employment and social benefits for relocated persons (v) The technical and managerial advantages of NIACL can be used to achieve quality development for the whole society (vi) More in line with the pattern of heterogeneous urbanization (vii) Must be led by higher levels of government | |

3.2.1. Compensation Scheme I: Direct Financial Compensation in Situ

This Scheme is a direct economic compensation scheme for the loss of NRACL benefits, mainly for the loss of benefits of the direct subject in CLR. CLR can significantly reduce the emission of pollutants, and the cultivated land or ecological land reclaimed in the CLR process will generate certain ecological benefits, which will help to achieve sustainable development [7]. In addition to compensation for economic losses, ecological compensation in the CLR process should also be increased. These compensations, which can be used for NRACL to improve their own development (e.g., improve access to education, improve housing conditions, improve living conditions, etc.), can in turn improve the living standards of residents. This is mainly achieved by increasing the compensation rate and expanding the scope of compensation. From the practice of direct economic compensation standards, (1) direct economic compensation has a hierarchy. First, at the municipal level, for the CLR project in the "198" area, the municipal government subsidizes the project at a subsidized price of 200,000 CNY/mu and coordinates about 25% of the quota for the municipal project. (2) The district government is responsible for 80% of the land grant for the CLR, and the municipal government is responsible for 20% of the land grant, so the district government bears the main cost of the CLR [34]. In Jinshan District, for example, the district government subsidizes with 1 million CNY/mu for the quota obtained through the reclamation of farmers' residential bases outside of the construction area, 700,000 CNY/mu for the quota obtained through the collation and reclamation of industrial and mining storage land outside of the construction area, and 350,000 CNY/mu for the quota obtained through the collation and reclamation of other construction land outside of the construction area. The quotas obtained from the CLR in Jinshan District, the district government coordinates about 80% for the protection of major projects at the municipal and district levels. [4] (3) In addition to the municipal and district subsidies, if it is not enough to cover the cost of the CLR, the shortfall will be borne by the township-level government; if there is a surplus of funds after deducting the cost of CLR, it will be used for the township to develop its own economy and improve the living conditions of the residents.

From the perspective of increasing the compensation rate, in the first round of the CLR "Three-year Action Plan" from 2014 to 2017, the municipal government's subsidy was 200,000 CNY/mu, while in the second round of CLR "Three-year Action Plan" from 2018 to 2020, the municipal government subsidy has been increased to 450,000 CNY/mu. This is an increase of 1.25 times. However, this is still far from enough to cover the cost of the CLR, and the subsidy rate needs to continue to be increased in the future. From the perspective of expanding the scope of compensation, the current subsidy rate of 450,000 CNY/mu should be increased, and the scope of the subsidy program should be expanded to cover the full scope of CLR projects, not just the "198" area.

In the early days, compensation was mainly for direct losses, which were not comprehensive, and such compensation was partial. With the improvement of the system, compensation for indirect losses was gradually reflected, but the proportion remained very low. There is a trend from partial to full compensation.

3.2.2. Compensation Scheme II: In Situ Enhancement of Development Capacity

This Scheme is an in situ development type program to enhance the in situ development capacity of the NRACL, mainly to increase the scale of development of NRACL's non-agricultural industries, upgrade the energy level of industries, and enhance the competitiveness of industries. It gives full play to NRACL's comparative advantages in terms of resource endowment, labor cost, ecological environment, etc., as well as its advantage of backwardness in terms of public service allocation, intelligent transportation construction, etc., to protect NRACL's development interests to a greater extent. NRACL's development requires certain construction land quotas, and after CLR, it leaves enough space for NRACL's development, leaving more quotas within the original property rights subject to industrial structure optimization and the development of public services, etc., to enhance the in situ development capacity of NRACL. Under the current conditions of industrial

concentration in the park and use control according to the plan, this Scheme can enable NRACL to improve its development capacity, realize its advantage of backwardness and maintain its long-term sustainable competitiveness. This type of compensation program benefits residents from this development by allocating more construction land quotas to NRACL for local economic development. As a result of the increased allocation of construction land targets and the increased efficiency of construction land, the government provides higher quality transportation, employment environment and other services.

### 3.2.3. Compensation Scheme III: Off-Site Enhancement of Development Capacity

This is an off-site development Scheme in which "people follow the quotas", i.e., the development quotas are used off-site, the corresponding employment is transferred, and the general population moves and becomes a member of the NIACL and enjoys the benefits of the NIACL. In other words, the outflow of the NRACL's construction land quotas should be accompanied by the outflow of a portion of the residents to the area where the construction land quotas are used, so as not to reduce or even increase the per capita welfare level of the remaining population in NRACL. Both the retained population and the outflow population can increase welfare and achieve the purpose of common development. CLR itself is the process of urbanization and the people's demand for employment is relatively high, and from the viewpoint of solving the quality of employment and improving employability, it is necessary to carry out the relocation of employment while the land quotas formed by CLR are used off-site. If this system is better organized, the employment pressure and demographic pressure of the population in NRACL will be transferred to NIACL with the relocation of the CLR quotas, so that the employment pressure and demographic pressure of the population remaining in NRACL will not increase or may even decrease, while allowing some of the people to enjoy better employment and living welfare in NIACL with the transfer of the construction land quotas. This model requires the establishment of a system for the relocation of people from the NRACL to the region where the reduction quotas are used, and the establishment of a sound supporting system by higher-level government departments. A similar feature is found in the cross-regional system of centralized housing and land acquisition now being implemented.

For the sake of illustration convenience, an example is given. Since specific data are difficult to measure, here are dummy data for the reader to understand the difference between compensation schemes II and III only. Assume that before the CLR, the economic development rate of both NIACL and NRACL was 6%. In fact, due to the location and technological advantages of NIACL, the economic development rate of NIACL is higher than that of NRACL, which is assumed to be equal for the sake of illustration. In compensation Scheme II, the construction land quota obtained by NRACL through CLR is used for the economic development of NRACL. All residents of NRACL benefit from this 8% economic development rate. In compensation Scheme III, NRACL's development rate decreases, for example, to 4%, because NRACL uses the construction land acquired through CLR off-site and only benefits from agricultural development. While the construction land quota is used for NIACL, due to the locational and technological advantages of NIACL, it leads to a higher economic development rate enhancement of NIACL, for example, from 6% to 10%. Residents who move from NRACL to NIACL can then enjoy the benefits of better economic development in the NIACL area. In fact, due to the locational and technological advantages of the NIACL, the economic growth rate of NIACL was greater than that of NRACL before the CLR, and thus the residents who transferred from NRACL to NIACL actually benefited more under compensation Scheme III. Residents who remain in NRACL, however, receive the benefit of only 4% of the economic growth rate. The specific benefit characteristics of these two scenarios are shown in Table 1.

Because of the different roles played by different levels of government in CLR, higher levels of government have more dominance over the construction land quotas obtained by CLR [14]. In the CLR process, different levels of government have different tendencies to allocate construction land quotas obtained by CLR, which affects the tendency of

government departments towards the three compensation schemes. In reality, the benefit coefficients for the townships and underlying subjects in the CLR area are below.

$$\lambda(r) = \frac{E(r)}{E(R)} \tag{1}$$

$$E(r) = f(r) \tag{2}$$

$$E(R) = F(R) \tag{3}$$

In Equations (1)–(3), $\lambda(r)$ is the benefit coefficient of the townships and the underlying subjects, $r$ is the use radius of land quotas obtained by CLR by the government departments at the township level and below, and $E(r)$ is the expected benefit to the government departments at the township level and below. $E(r)$ is the logistic function of $r$, which is in the increasing stage at this stage. $R$ is the use radius of land quotas obtained by CLR by government departments at the district level and above, and $E(R)$ is the expected gain for government departments at the district level and above. $E(R)$ is the logistic function of $R$, which is in the increasing stage at this stage. The schematic diagram is shown in Figure 2. The horizontal axis represents the use radius of the land quotas obtained by the CLR, $ME$ and $AE$, which denote the marginal and average benefits corresponding to each use radius, respectively. The benefit coefficients for government departments at the district level and above in the CLR area are below.

$$\lambda(R) = 1 - \lambda(r) \tag{4}$$

In Equation (4), the $\lambda(R)$ is the benefit factor for government departments at the district level and above.

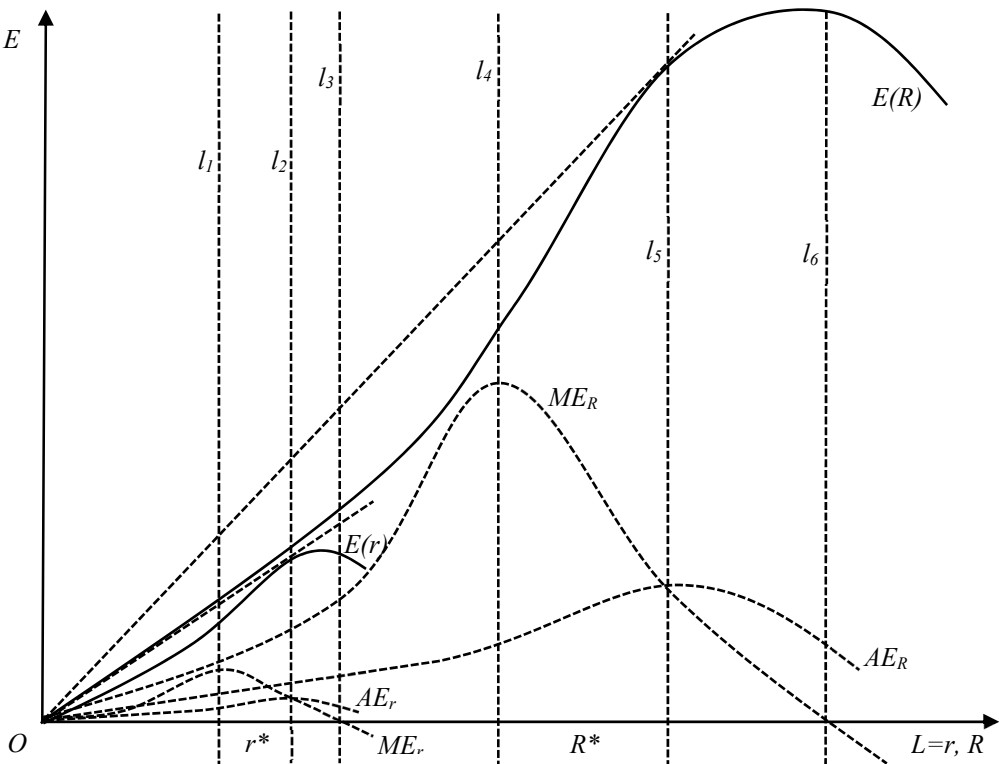

**Figure 2.** Schematic diagram of the construction land allocation in the CLR process.

From the research: (1) The radius of transactions chosen by municipal and district government departments $E(R)$ is gradually growing. This is because the increase in $R$

results in greater value-added potential and greater net benefits for construction land. (2) The grassroots government and the residents of NRACL are more inclined to increase the compensation standard or expand the scope of compensation, followed by in situ and off-site enhancement of development capacity. (3) From Figure 2, we can find that the radius $r^*$ of optimal allocation of land quotas by government departments at the township level and below lies between $l_1$ and $l_2$. Figure 2 shows that the optimal radius $R^*$ of land allocation for government departments of municipal districts lies between $l_4$ and $l_5$. Thus, it appears that the higher-level government wants to expand the radius of CLR to optimize the allocation of construction land quotas so as to achieve sustainable development, while the net reduction subject opposes expanding the radius of CLR because the benefit radius is not large; thus, a contradiction between the two emerges. (4) In order to resolve the contradiction arising in (3), the lower-level government tries to expand the radius of benefit $r$ as much as possible.

At this stage, it is becoming more and more expensive for the government to carry out CLR and based on the principle of maximizing returns, there is a need to expand the radius of trading of quotas. The institutional problem is to solve the problem of employment placement and compensation in the workforce. The closer $r$ is to $R$, the smaller the frictional cost of the CLR will be, and that is when a higher-level, systemic policy for the benefits of the CLR will be needed.

### 3.3. Factors Influencing the Behaviour of Residents in Choosing between the Three Compensation Schemes

Theoretically, each of the three compensation schemes has advantages and disadvantages, and residents' choice of these compensation schemes is influenced by a variety of factors at the macro and micro levels.

#### 3.3.1. Macro Factors

(1)   Development Orientation of Municipality to District

CLR is an innovative mechanism to seek continued development through the spatial release of construction land under the control of the total amount and intensity of construction land and is a reallocation of land development rights under planning constraints. Through the previous analysis, under the planned land use control, the allocation of construction land is in the form of "reduction to increase". The allocation of land quotas obtained by the CLR is mainly coordinated at the district level, and the townships have limited land quotas to use. The extent to which "reduction for increase" is achieved at the district level is stronger.

The development orientation of the municipality to the district is reflected in two main aspects: first, for NIACL, there are often more municipal projects and thus more development opportunities using municipal quotas; second, a small reduction task and more quotas are used in the CLR process. In the CLR process, if a district's CLR results in a larger proportion of land quotas for new construction land, the more the municipality positions the district as a NIACL, the district is a reserved or development area, and the construction land quotas are not reduced. This is beneficial for the development of the district and the residents of the district can benefit from the development. At this stage, if the CLR of a district forms a larger proportion of land quotas for new construction land, the larger proportion of quotas after the demolition of poorly efficient enterprises is still used as construction land in the district, which falls under the category of construction land renewal. Compared to NRACL, the residents of the district are likely to focus on the renewal of construction land and pay less attention to the compensation scheme of CLR. Based on the above analysis, the following hypothesis is put forward:

**Hypothesis 1 (H1):** *Residents in the construction land renewal area are more concerned with regional renewal and less concerned with various compensation schemes.*

(2)  Reasonableness of Compensation Standard

CLR is a dynamic process and the compensation rate for CLR increases with economic development, and the better the economic conditions, the higher the direct compensation rate. This will affect residents' expectations of CLR compensation standards, which in turn affects residents' expectations of compensation benefits. Benefits expectation is an important factor influencing residents' choice behavior [35]. If the compensation standard at the current stage is more reasonable, then the more optimistic residents' expectations of future compensation will be. It should be noted that the compensation in Scheme I is for the future compensation scheme, which is an unimplemented scheme scenario based on the policy practice. The compensation standard here is the reasonableness of the current compensation standard, and the subjective perception of the current compensation standard of the residents is used in the empirical research. The more reasonable the compensation standard at the current stage, the fewer the residents who will lose in the CLR process and thus may be more favorable to both in situ and off-site enhancements of development capacity. Based on the above analysis, the following hypothesis is put forward:

**Hypothesis 2 (H2):** *The more reasonable the compensation standard in the CLR process, the stronger the preference of residents for schemes I, II and III.*

(3)  Employment Pressure on Township

The CLR process is accompanied by the closure and dismantling of inefficient enterprises, which may have a negative impact on employment in NRACL. In turn, the principle of allocation of efficiency priority in the allocation of construction land quotas leads to the underallocation of construction land quotas in NRACL, which may have some impact on employment. As a result, residents are increasingly concerned about employment solutions rather than one-time financial compensation. Both in situ and off-site enhancements of development capacity contribute to employment expansion and thus may be preferred by residents. Based on the above analysis, the following hypothesis is put forward:

**Hypothesis 3 (H3):** *The higher the employment pressure in the township, the stronger the preference of residents for schemes II and III, and the lower the preference for Scheme I.*

(4)  Location Conditions

Location is an important factor affecting CLR. Location conditions affect both the compensation criteria of CLR and the quota allocation after CLR. The reduction of CLR is based on location disadvantage, while the quota allocation is based on location advantage [8]. NRACL has a location disadvantage, and the residents of this area have lower expectations for economic compensation. Location disadvantage is detrimental to the long-term development of NRACL, and there is an urgent need to improve the competitiveness of the region and expand employment and improve the welfare of residents. As a result, districts with poor locations are more desirous of development and have a greater preference for both in situ and off-site enhancements of development capacity, and a weaker preference for one-time financial compensation. Based on the above analysis, the following hypothesis is put forward:

**Hypothesis 4 (H4):** *The worse the locational conditions, the lower the preference of residents for Scheme I and the higher the preference for schemes II and III.*

(5)  Population Density of Township

The higher the population density, the greater the need for a sufficient rate of economic development to address the basic living, employment and development needs of residents. Concentrated housing helps to reduce the waste of land for rural construction [36]. The higher the population density, the higher the demand of residents for better livelihoods and employment, and therefore the greater the incentive for development. The higher the

population density, the wider the benefits involved in compensation and the higher the cost of carrying out CLR. Based on the above analysis, the following hypothesis is put forward:

**Hypothesis 5 (H5):** *The higher the population density of the area, the stronger the preference of residents for schemes I, II and III.*

(6)　Development Pressure on Township

CLR is a way of pursuing high-quality development under planning land use controls. The greater the development pressure, the greater the need to enhance the employment capacity of the region and its capacity for sustainable development. Under the "reduce to increase" quotas use mechanism, it is more necessary to use CLR to free up construction land quotas to achieve the purpose of development. Based on the above analysis, the following hypothesis is put forward:

**Hypothesis 6 (H6):** *The greater the development pressure in the area, the stronger the preference of residents for Scheme I, Scheme II and Scheme III.*

3.3.2. Micro Factors

The variability of residents' individual and household characteristics may influence their choice of the three compensation schemes. These factors include individual characteristics such as residents' gender, age, and education level, as well as household characteristics such as residents' household income and household size structure. The resettlement of residents in centralized residential areas through the reduction of the homestead is an important way to save land for construction. However, the limited capacity of older groups leads to a reduced ability to cope with environmental stresses caused by external changes [36]. As a result, older residents may be more inclined to receive financial compensation. Residents with higher levels of education and higher household income are more competitive, and CLR exerts less of a negative impact on them. However, due to their high competitiveness, they may have a greater preference for Scheme II which enhances the competitiveness of the region. Residents with higher family dependency ratios have a higher need for quality education or health care, and thus may have a greater preference for Scheme III. Based on the above analysis, the following hypothesis is put forward:

**Hypothesis 7 (H7):** *Individual resident characteristics and household characteristics can affect residents' compensation scheme selection behavior to varying degrees. Compared to females, males have a lower preference for all three schemes, especially compensation Scheme III. Older residents may prefer compensation Scheme I. Residents with higher education levels may prefer compensation Scheme II and have a lower preference for other compensation schemes. Residents with higher household incomes may prefer all three schemes. Residents with higher family dependency ratios are likely to have a greater preference for compensation Scheme III. Based on the above theoretical analysis, a theoretical analysis framework applicable to the analysis of residents' choices of different compensation schemes is established in conjunction with the compensation characteristics of CLR, as shown in Figure 3.*

Table 2 is a generalized presentation of the research hypotheses presented in this paper.

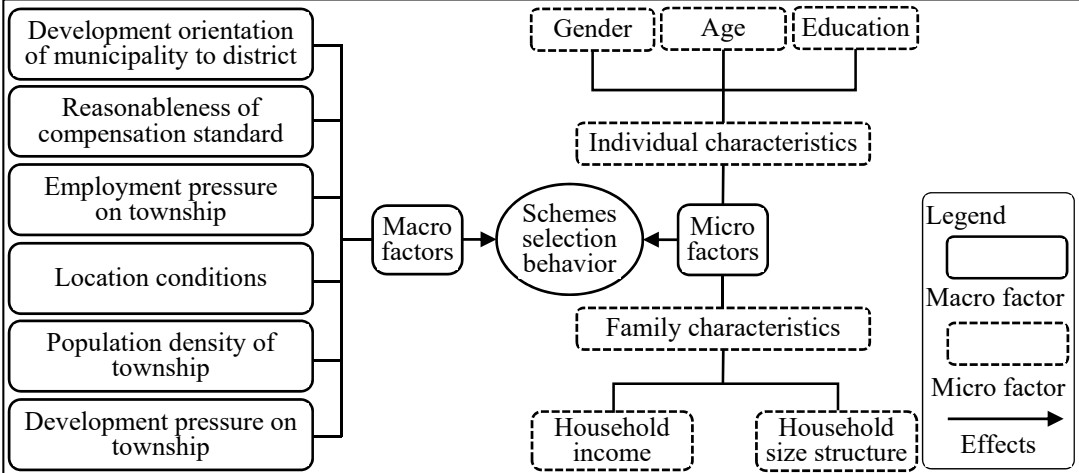

**Figure 3.** The theoretical analysis framework of this paper.

**Table 2.** A generalized presentation of the research hypotheses.

| Influencing Factor | Specific Influencing Factor | Hypothesis | Scheme I | Scheme II | Scheme III |
|---|---|---|---|---|---|
| | Development orientation of municipality to district | H1 | − | − | − |
| | Reasonableness of compensation standard | H2 | + | + | + |
| Macro factors | Employment pressure on township | H3 | − | + | + |
| | Location conditions | H4 | − | + | + |
| | Population density of township | H5 | + | + | + |
| | Development pressure on township | H6 | + | + | + |
| | Gender | | − | − | − |
| | Age | | + | +/− | +/− |
| Micro factors | Level of education | H7 | − | + | − |
| | Household income | | + | + | + |
| | Household size structure | | +/− | +/− | + |

Note: (1) "+" represents positive influence, i.e., preference, "−" represents negative influence, i.e., not preference, and "+/−" represents uncertainty influence. (2) In the empirical part, location conditions are further subdivided into village location and town location. The impact here is exemplified by the locational disadvantage.

## 4. Research Methodology and Data

### 4.1. Model Setting

In the CLR process, to compensate for the losses caused by the net reduction in construction land on the direct subject of CLR, there are three compensation schemes for residents to choose from: Scheme I, Scheme II and Scheme III. Residents' choices of these compensation schemes are somewhat correlated, but not mutually exclusive. Although these compensation schemes are in general characterized by a reciprocal nature, some of the specific measures of these compensation schemes can be carried out simultaneously. In reality, there are also different models being designed. Thus, a multi-objective empirical study can be conducted.

In multi-objective decision making, the choices between sources of information are not mutually exclusive, making it possible for the random error terms to be correlated with each other [37]. Considering the correlation of residents' preferences for the choice of compensation schemes, a joint estimation of the choice of the three compensation schemes is required. The use of regression models such as simple binomial logit or multinomial logit does not allow for addressing multiple dependent variables and is not conducive to cross-sectional comparisons for multi-objective decision making. In contrast, the MVP model allows for correlation between the error terms of different equations [37,38]. Therefore, this paper uses the MVP model, which can handle multiple binary choices simultaneously, to

study the behavior of residents' choices of different compensation schemes. Drawing on related research [39], an MVP model of the following form is set:

$$y_{im}{}^* = \beta_m{}'X_{im} + \varepsilon_{im}, \ m = 1, \ldots, M \tag{5}$$

In Equation (5), the dependent variables satisfy:

$$y_{im} = \begin{cases} 1 \ if \ y_{im}{}^* > 0 \\ 0 \ otherwise \end{cases} \tag{6}$$

In Equations (5) and (6), $i$ represents the $i$ resident, and $M$ is the number of programs. While $\varepsilon_{im}$ ($m = 1, \ldots, M$) is the random error term that obeys multivariate normal distribution, $y_{im} = 1$ and $y_{im} = 0$ denote the number of $i$, respectively. The resident chooses and does not choose the $m$ compensation scheme. $X_{im}$ is the set of influencing factors that affect residents' choice of compensation schemes. $X_{im}$ includes the development orientation of the municipality to the district, the reasonableness of compensation standard, the employment pressure on the township, the location conditions of the township, the location conditions of the village, the population density of the township, the development pressure on the township, the gender, age as well as the education level of the residents, and their household income and household size structure.

*4.2. Selection of Variables and Indicator Measures*

4.2.1. Dependent Variables

The dependent variables are residents' preferences for choosing among the three compensation schemes for CLR. These include residents' preferences for Scheme I ($Y_1$), Scheme II ($Y_2$) and Scheme III ($Y_3$).

4.2.2. Explanatory Variables

The explanatory variables include both macro- and micro-level variables: (1) Macro variables. The development orientation of the municipality to the district ($DO$), the reasonableness of the compensation standard ($CS$), the employment pressure of the township ($EP$), the location condition of the township ($LCT$), the location condition of the village ($LCV$), the population density of the township ($PD$), the development pressure of the township ($DP$), the gender, age, education level and residents' household income and household size structure. (2) Micro variables. There are many factors that affect the off-site employment of NRACL residents. Working conditions are a complex matter that cannot yet be investigated in detail in this manuscript. In the compensation Scheme III, due to the limited number of relocations, the relocated residents are relatively high-quality residents, and we assume that the residents are in good working condition. Additionally, we chose the family dependency ratio as a variable that affects the work of the population. These micro variables include individual characteristics of residents and household characteristics variables. Individual characteristics variables include residents' gender ($GEN$), age ($AGE$) and education level ($EDU$). Household characteristics include household income ($HI$) and household size structure ($HSS$).

Considering the heterogeneity of resident status, the resident status variable ($GB$) is added for heterogeneity analysis. Since the use of construction land is also affected by the types of land use planning in the CLR process, the influence of the types of land use planning is also considered in this paper. For the types of land use planning, the study area is divided into three types of areas based on the increase or decrease of the planned construction land area compared with the current construction land area. The current construction land area is the construction land area in 2016. The planned construction land area is the construction land area in 2035. The first type is the change of the average planned construction land area between 10% and 50%, which is called the planning increment type area ($U$). The second type is the change of the average planned construction land area between −10% and 10%, which is called the planning balance type area ($V$). The third type

is the change of the average planned construction land area between −10% and −50%, which is called the planning reduction type area (*Z*). In the model, the *Z* group is the reference for regression analysis. The specific interpretation and indicator measures of each variable of the model are shown in Table 3.

**Table 3.** Description of model variables.

| Variable Type | Variable Name | Variable Code | Description |
|---|---|---|---|
| Dependent variables | Preferences of Scheme I | $Y_1$ | Options for increasing the price of the use of the quota or expanding the scope and intensity of compensation: "Yes" = 1, "No" = 0 |
| | Preferences of Scheme II | $Y_2$ | Options for appropriately increasing the use of construction land quotas in the area being reduced or improving the competitiveness of the region's industries through financial and technical support: "Yes" = 1; "No" = 0 |
| | Preferences of Scheme III | $Y_3$ | Options for enhancing the transfer of the remaining rural population to areas where construction land quotas are used: "Yes" = 1; "No" = 0 |
| Explanatory variables | Development orientation of municipality to district | $DO$ (%) | The total area of new construction land required within the development boundary for each district from the planning base year to 2035/the total area of reduced construction land required outside the development boundary |
| | Reasonableness of compensation standard | $CS$ | Evaluation of the reasonableness of the reduced compensation standard in this township compared to other townships: "Very reasonable" = 5; "Quite reasonable" = 4; "Average" = 3; "Quite unreasonable" = 2; "Very unreasonable" = 1 |
| | Employment pressure on township | $EP$ | Rating of how negatively you and your family's job opportunities have been affected by reduction planning and policies: "Very little" = 1; "Quite little" = 2; "Average" = 3. "Quite large" = 4; "very large" = 5 |
| | Location condition of township | $LnLCT$ (kilometres) | Logarithm of the distance from the town to the district government station |
| | Location condition of village | $LnLCV$ (kilometres) | Logarithmic value of the distance from the village to the township government station |
| | Population density of township | $LnPD$ (persons/square kilometers) | Logarithmic value of the resident population of the township per unit of administrative area |
| | Development pressure on township | $DP$ (%) | Ratio of the total value of industrial output on the scale in the first year of the 14th Five-Year Plan to the first year of the 13th Five-Year Plan in each township |
| | Gender | $GEN$ | "Male" = 1, "Female" = 0 |
| | Age | $AGE$ | "30 years and below" = 1; "31–45 years" = 2; "46–59 years" = 3; "60 and above" = 4 |
| | Level of education | $EDU$ | "Primary school and below" = 1; "Lower secondary school" = 2; "Upper secondary school" = 3; "College and above" = 4 |
| | Household income | $HI$ | "50,000 CNY and below" = 1; "50,000 CNY–100,000 CNY" = 2; "100,000 CNY–200,000 CNY" = 3; "200,000 CNY and above" = 4 |
| | Household size structure | $HSS$ (%) | Household dependency ratio |
| Heterogeneous variables | Resident status | $GB$ | Village cadres, township cadres and above = 1; others = 0 |
| | Types of land use planning | Is it a U region ? Is it a V region ? | "Yes" = 1, "No" = 0 "Yes" = 1, "No" = 0 |

### 4.3. Data Sources and Descriptive Statistics

CLR in Shanghai is implemented in a basic unit of village committees. The government compensation for the reduction of scattered construction land on the village committee and the concentration on the construction area is also carried out with the village committee as the basic unit. The research group conducted data collection through interviews and questionnaire distribution. In order to increase the representativeness of the sample, the subject group selected Y District, W District and X District in Shanghai as the research sites according to the economic development conditions and location characteristics, covering all characteristic types of CLR in Shanghai, China. In terms of sampling method, it drew on established studies using the survey method of spatial episodic rather than random sampling, which is an alternative method to random sampling in academic studies when there are objective difficulties in random sampling [40]. A total of 2400 questionnaires were distributed, and 2354 questionnaires were returned, and after excluding samples with

missing information, inconsistencies and outliers, 2192 valid questionnaires were finally obtained, with a valid return rate of 93.12%. The survey period is from March to May 2021. The macro statistics in this paper are the data of 28 townships in the Y District, W District and X District, and the data are obtained from the statistical yearbook of each district. The distance data are based on the administrative division map of each district and obtained using the Near analysis tool in the Arctoolbox in ArcGIS.

The data used to classify the types of land use planning were obtained from the *Master Plan and General Land Use Plan of Y District, Shanghai (2017–2035)*, *Master Plan and General Land Use Plan of W District, Shanghai (2017–2035)*, and *Master Plan and General Land Use Plan of X District, Shanghai (2017–2035)*.

The descriptive statistics of the variables are presented in Table 4. The preference of the residents for the choice of compensation schemes for CLR is presented in Figure 4. As seen in Figure 4, the respondents have the strongest preference for Scheme I, followed by a preference for Scheme III and the lowest preference for Scheme III. This indicates that, at this stage, residents are more concerned with direct compensation for economic losses than in situ and off-site enhancements of development capacity.

**Table 4.** Descriptive statistics of the variables.

| Variable | Obs | Mean | Std. Dev. | Min | Max |
|:---:|:---:|:---:|:---:|:---:|:---:|
| $Y_1$ | 2192 | 0.8335 | 0.3726 | 0.0000 | 1.0000 |
| $Y_2$ | 2192 | 0.4676 | 0.4991 | 0.0000 | 1.0000 |
| $Y_3$ | 2192 | 0.5192 | 0.4997 | 0.0000 | 1.0000 |
| $DO$ | 2192 | 61.7834 | 17.0095 | 53.8462 | 99.7349 |
| $CS$ | 2192 | 4.0087 | 0.8516 | 1.0000 | 5.0000 |
| $EP$ | 2192 | 1.8125 | 0.8829 | 1.0000 | 5.0000 |
| $LnLCT$ | 2192 | 2.4523 | 1.1146 | −0.6280 | 3.6226 |
| $LnLCV$ | 2192 | 0.9412 | 0.9167 | −9.9443 | 2.5221 |
| $LnPD$ | 2192 | 6.5469 | 0.3757 | 5.3628 | 7.5391 |
| $DP$ | 2192 | 106.0134 | 34.1628 | 59.0644 | 183.0224 |
| $GEN$ | 2192 | 0.5584 | 0.4967 | 0.0000 | 1.0000 |
| $AGE$ | 2192 | 2.5032 | 0.9390 | 1.0000 | 4.0000 |
| $EDU$ | 2192 | 3.0780 | 1.0478 | 1.0000 | 4.0000 |
| $HI$ | 2192 | 2.6428 | 1.0186 | 1.0000 | 4.0000 |
| $HSS$ | 2192 | 36.2963 | 33.9932 | 0.0000 | 100.0000 |
| $GB$ | 2192 | 0.2778 | 0.4480 | 0.0000 | 1.0000 |
| $U$ | 2192 | 0.0899 | 0.2861 | 0.0000 | 1.0000 |
| $V$ | 2192 | 0.1683 | 0.3743 | 0.0000 | 1.0000 |

Note: A family dependency ratio of 1 is due to the fact that the family is entirely composed of people in need of support.

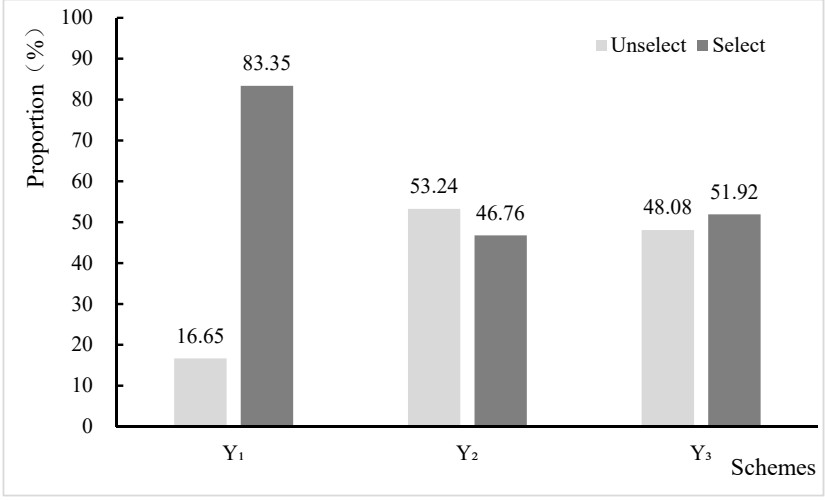

**Figure 4.** Residents' preferences for multiple compensation schemes for CLR.

## 5. Empirical Results and Analysis

### 5.1. Baseline Regression Results

By performing a maximum likelihood estimation of Equation (5) [37], the estimated values of $\beta$ can be obtained. The baseline regression results of the MVP model in this paper are presented in Table 5. Referring to the principle of setting the number of random draws slightly greater than the arithmetic square root of the sample size for robust regression results in the empirical estimation of the MVP model [41], the number of draws was set to 50. The regression likelihood ratio test for the dependent variable of the MVP model showed a chi-square value of rho21 = rho31 = rho32 = 76.1670 with a significance probability *p*-value of 0.0000. Thus, the original hypothesis that the correlation coefficient of the dependent variable is 0 is rejected, indicating the validity of using the MVP model.

**Table 5.** Baseline regression results.

| Variable | (1) $Y_1$ | (2) $Y_2$ | (3) $Y_3$ |
|---|---|---|---|
| DO | −0.0030 (0.0020) | −0.0013 (0.0018) | −0.0022 (0.0017) |
| CS | 0.2104 *** (0.0424) | 0.0248 (0.0359) | 0.0673 * (0.0360) |
| EP | −0.2047 *** (0.0395) | 0.0300 (0.0349) | 0.0641 * (0.0352) |
| LnLCT | −0.1153 *** (0.0386) | 0.1448 *** (0.0329) | 0.1507 *** (0.0324) |
| LnLCV | −0.0290 (0.0347) | 0.0022 (0.0297) | −0.0243 (0.0295) |
| LnPD | 0.1991 * (0.1158) | 0.3359 *** (0.0972) | 0.0803 (0.0943) |
| DP | 0.0020 * (0.0011) | 0.0021 ** (0.0009) | 0.0008 (0.0009) |
| GEN | −0.0345 (0.0662) | −0.0559 (0.0548) | −0.1034 * (0.0547) |
| AGE | 0.1485 *** (0.0452) | 0.0556 (0.0380) | 0.0127 (0.0383) |
| EDU | −0.0926 ** (0.0452) | 0.1323 *** (0.0367) | −0.0422 (0.0368) |
| HI | 0.1046 ** (0.0413) | 0.1052 *** (0.0348) | 0.1330 *** (0.0347) |
| HSS | 0.0003 (0.0011) | 0.0009 (0.0009) | 0.0016 * (0.0009) |
| Constant | −0.8146 (0.9091) | −3.7602 *** (0.7757) | −1.4066 * (0.7575) |
| Draws | | 50 | |
| Atrho21 | | −0.2228 *** (0.0400) | |
| Atrho31 | | −0.2291 *** (0.0407) | |
| Atrho32 | | 0.1742 *** (0.0343) | |
| Wald | | 240.55 *** | |
| Likelihood ratio test | | 76.1670 *** | |
| Observations | | 2192 | |

Note: Robust standard errors in parentheses, ***, ** and * indicate significance at the 1%, 5% and 10% levels, respectively.

5.1.1. Macro-Influencing Factors

(1) The development orientation of the municipality to the district is not significant. The near-term plan is 2020–2025; the long-term plan is 2025–2035. The long-term plan has not yet been implemented. At this stage, the construction land used by the districts for development comes from the use of land quotas obtained from CLR for more efficient areas and from the renewal of the district's stock of construction land, i.e., the upgrading of the capacity of the stock of construction land. Rather than CLR dominating economically developed areas in general, the pressure and incentive to renew construction land dominates. The existing system stipulates that 25% of the quotas after CLR in each district go to the municipal level, and the rest all goes to the districts for their own use. If the districts want more construction land quotas, they need to do more CLR. Such a provision is in conflict with the district-level development orientation. A coordination mechanism would be to plan for NIACL by increasing the CLR tasks and thus gaining more construction land use quotas. As shown in Figure 1, under the planned land use control, the allocation of construction land is "reduced to increase", and the allocation of land quotas obtained from CLR is mainly coordinated within the district. The "reduction for increase" is realized to a greater extent at the district level. Therefore, the greater the proportion of the CLR-derived land quotas used for new construction land in the district, the more likely it is that the district as a whole is a construction land regeneration area so that residents are more concerned about the renewal of construction land in the district and less concerned about the CLR, which is probably why the development orientation of the municipality to the district does not have a significant influence on residents' choice of schemes. Hypothesis 1 (H1) was verified. This is consistent with the relevant research on urban renewal [10,42,43]. Urban renewal has been an important strategic choice to promote urban development [10]. In contrast to NRACL, NIACL needs to address development challenges through urban renewal. There is a lack of attention to the reduction of construction land and its compensation.

(2) Reasonableness of compensation standards significantly enhances residents' choice of schemes I and III. Reasonableness of compensation standards helps to prompt residents to choose Scheme II, but this effect does not pass the 10% significance level test, which may be due to the fact that enhancing local competitiveness relies more on technical help from economically developed regions and the role of economic compensation is limited. From the current stage of CLR, the compensation to the relevant interest subjects is mainly economic compensation, and in the absence of other compensation, residents attach more importance to economic compensation. Thus, the more reasonable the compensation standard, the greater the likelihood that residents will benefit from the compensation, and thus this increases the number of residents who have a preference for Scheme I. If the compensation is more reasonable, then the residents expect more off-site employment, off-site development and improved living conditions, and thus the reasonableness of the compensation increases the residents' preference for Scheme III. Hypothesis 2 (H2) was verified. As with the existing related studies, the loss of development in NRACL due to CLR needs to be compensated by reasonable compensation [2,9,34,35].

(3) The employment pressure of the township significantly increases the probability of residents choosing Scheme III. CLR is accompanied by the closure of inefficient enterprises, and under the "reduce to increase" land allocation rule, NRACL are at a disadvantage in the process of adding new construction land quotas, making it difficult to introduce high-quality enterprises, and employment pressure becomes a real problem [2,9,14]. As a result, the greater the employment pressure, the greater the preference of residents for off-site schemes to enhance development capacity. Employment pressure in the township also increases the probability of residents choosing Scheme II, but this effect does not pass the 10% level of the significance test. The possible reason for this is that while enhancing development capacity through the

region also helps to expand employment, this route is slow and long-lasting, and thus residents' preference for it is not significant. Employment pressure in the township significantly reduces the probability of residents choosing Scheme I, suggesting that employment pressure reinforces residents' concerns about employment. Hypothesis 3 (H3) was verified.

(4) The locational disadvantage of the township significantly increases the probability that residents will choose schemes II and III. The district can choose either a net increase or a net decrease in construction land quotas in the allocation of land quotas formed by the CLR, which depends on the district's development orientation of the township [8]. This development orientation is influenced by the locational conditions of the township. The further the township is from the district administrative center, the less potential it has for allocating construction land quotas, and the economic development effect of using the land quotas formed by it for a township with a better location is better than the allocation of land quotas in a disadvantaged location. That is, the larger the allocation radius of the land quotas obtained by CLR, the better the benefits of the allocation. Moreover, the locality disadvantaged areas need technical support from developed areas to improve their competitiveness. Thus, the residents of areas with poor locations in the townships have a strong desire to develop, either locally or off-site, to enhance their development capacity.

The location disadvantage of the township significantly reduces the probability of the residents choosing Scheme I. This is due to the fact that the residents of poorly located townships are themselves economically weak and thus have low expectations of the potential for growth in economic compensation standards. They have low expectations of the potential for future compensation increases, and they have higher expectations for development or relocation. From the research, the residents of poorly located townships have a stronger need to increase the scale of development of non-agricultural industries in the CLR region, to improve the capacity and competitiveness of industries and to pay more attention to the improvement of living conditions. As a result, residents of poorly located townships have a higher preference for Scheme II and Scheme III.

(5) The influence of village location conditions on residents' choice of Scheme I, Scheme II and Scheme III is not significant. The possible reasons for this are as follows: since the land quotas formed by CLR are mainly allocated at the township level [8], villages lack bargaining power in terms of quota acquisition [2,9,14], and since townships in Shanghai are generally small and have convenient transportation conditions, the distance between villages is not a major factor of influence; thus, the effect of village location on residents' choice behavior is not significant. Hypothesis 4 (H4) was verified.

(6) The population density of townships significantly increases the probability that residents will choose Scheme I and Scheme II. Townships with high population densities also tend to be reserved or development areas, which are also better developed in their own right and have a cumulative effect of development over the years and have an agglomeration advantage. Residents in these areas have an incentive to improve their livelihoods and employment, and thus have a preference for Schemes I and II. In addition, since the higher the population density of the area, the greater the resistance to population migration in the CLR process, the residents have less opportunity to benefit from Scheme III compared to schemes I and II, and thus the residents are less dependent on Scheme III. Hypothesis 5 (H5) was verified.

(7) The higher the development pressure, the more the township tends to develop itself and increase its income level [8,9,11], with less support for cross-regional development. The greater the development pressure, the more the industry has to upgrade, and CLR can increase the mobility of enterprises so that backward enterprises exit and new enterprises enter. As a result, residents have a greater preference for schemes I and II. Due to development pressures, residents have a weaker preference for off-site upgrading of development capacity, preferring either financial compensation or the

upgrading of local development capacity for long-term development. Thus, there is a weaker preference for Scheme III. Hypothesis 6 (H6) was verified.

### 5.1.2. Micro-Influencing Factors

(8)  Men have a lower preference for Scheme III compared to women. This may be due to the comparative advantage of men in the job market, where men have a weaker preference than women for compensation for off-site enhancement of development capacity.

(9)  The older the resident, the more he/she prefers Scheme I. The older the resident, the more he/she prefers direct financial compensation that can directly improve his/her living conditions (this is consistent with existing studies) [36].

(10)  Residents with higher education levels show a higher preference for Scheme II and a lower preference for Scheme I. The more educated residents are more aware of CLR and more receptive to compensation schemes that enhance NRACL's ability to develop in situ. Thus, they do not prefer short-term, one-time compensation schemes for economic losses. The effect of this on their preference for Scheme III is not significant.

(11)  Households with higher household incomes have higher expectations of economic development and thus have a higher preference for all three available schemes.

(12)  Residents with a higher proportion of dependent family members have a higher preference for Scheme III. A high proportion of family dependents indicates a high number of children attending school or elderly people and therefore a high demand for quality education or health care. The use of the construction land quota in a different location, along with the corresponding movement of employed persons and the general population and the consequent benefits of developed areas, including employment, health care and education, is attractive to families with high family support pressure. Hypothesis 7 (H7) was verified.

### 5.2. Heterogeneity Regression Results

#### 5.2.1. Heterogeneity of Resident Status

Considering the cadre heterogeneity, this paper further includes cadre dummy variables for the heterogeneity test. To facilitate a comparison of the significance of the coefficients, we examine the effect of heterogeneity on resident status by including dummy variables. The results are presented in Table 6. The MVP model is empirically estimated by setting the number of random draws to 50. The MVP model regression likelihood ratio test shows a chi-square value of 76.2250 for rho21 = rho31 = rho32 = 0 and a significance probability *p*-value of 0.0000. Thus, the rejection of the original hypothesis that the correlation coefficient of the dependent variable is 0, indicating the validity of using the MVP model.

From Table 6, it can be found that (1) the baseline regression results of this paper are robust. (2) The cadres prefer Scheme I and Scheme III more, while their preference for Scheme II is lower, but it does not pass the 10% significance level test. Possible reasons for this are as follows: At this stage, the performance appraisal of the higher level government to the lower level government includes both the GDP appraisal of the year and the appraisal of the CLR's task completion. The appraisal is a bottom line and an incentive for cadres, reflecting the government's development strategy from top to bottom. Of these two appraisal objectives, the appraisal of GDP is relatively simple and straightforward. (1) When the two goals of completing GDP and CLR tasks are in conflict, the completion of CLR tasks is well manipulated. Currently, the pressure from the CLR assessment exceeds the pressure on GDP growth. (2) NRACL's own growth potential is getting smaller due to the transfer of construction land quotas to NIACL, which is one of the reasons why cadres are not active in increasing GDP through Scheme II. (3) Increasing direct financial compensation to the subject of reduction is a way to increase residents' income, which is a performance for the cadres, and the increase in residents' income also helps the CLR task because the increase in residents' income makes them more supportive of CLR and reduces the resistance to its implementation. (4) Due to research difficulties, municipal cadres and

district cadres were not surveyed, which may also be a reason for the insignificant impact of the cadres in this paper.

**Table 6.** Heterogeneity regression results of resident status' impact.

| Variable | (1) | (2) | (3) |
|---|---|---|---|
| | $Y_1$ | $Y_2$ | $Y_3$ |
| GB | 0.1256 | −0.0275 | 0.0139 |
| | (0.0858) | (0.0701) | (0.0700) |
| DO | −0.0023 | −0.0015 | −0.0022 |
| | (0.0021) | (0.0018) | (0.0018) |
| CS | 0.2078 *** | 0.0253 | 0.0670 * |
| | (0.0425) | (0.0359) | (0.0360) |
| EP | −0.2004 *** | 0.0288 | 0.0646 * |
| | (0.0396) | (0.0350) | (0.0354) |
| LnLCT | −0.1147 *** | 0.1446 *** | 0.1508 *** |
| | (0.0386) | (0.0329) | (0.0324) |
| LnLCV | −0.0233 | 0.0010 | −0.0237 |
| | (0.0349) | (0.0298) | (0.0296) |
| LnPD | 0.2088 * | 0.3346 *** | 0.0809 |
| | (0.1155) | (0.0973) | (0.0943) |
| DP | 0.0021 ** | 0.0020** | 0.0008 |
| | (0.0011) | (0.0009) | (0.0009) |
| GEN | −0.0402 | −0.0548 | −0.1041 * |
| | (0.0661) | (0.0548) | (0.0548) |
| AGE | 0.1448 *** | 0.0562 | 0.0125 |
| | (0.0453) | (0.0381) | (0.0383) |
| EDU | −0.1115 ** | 0.1365 *** | −0.0443 |
| | (0.0470) | (0.0384) | (0.0384) |
| HI | 0.0958 ** | 0.1072 *** | 0.1321 *** |
| | (0.0417) | (0.0352) | (0.0350) |
| HSS | 0.0003 | 0.0009 | 0.0016 * |
| | (0.0011) | (0.0009) | (0.0009) |
| Constant | −0.8781 | −3.7509 *** | −1.4104 * |
| | (0.9063) | (0.7760) | (0.7578) |
| Draws | 50 | | |
| Atrho21 | −0.2225 *** | | |
| | (0.0400) | | |
| Atrho31 | −0.2297 *** | | |
| | (0.0407) | | |
| Atrho32 | 0.1742 *** | | |
| | (0.0343) | | |
| Wald | 242.42 *** | | |
| Likelihood ratio test | 76.2250 *** | | |
| Observations | 2192 | | |

Note: Robust standard errors in parentheses, ***, ** and * indicate significance at the 1%, 5% and 10% levels, respectively.

5.2.2. Heterogeneity of the Types of Land Use Planning

Considering the heterogeneity of the types of land use planning, this paper further introduces a dummy variable for the types of land use planning and conducts a heterogeneity test. To facilitate a comparison of the significance of the coefficients, we examine the effect of heterogeneity on the types of land use planning by including dummy variables. The results are presented in Table 7. The number of random draws is set to 50 for the empirical estimation of the MVP model. The MVP model regression likelihood ratio test shows a chi-square value of 74.8275 for rho21 = rho31 = rho32 = 0 and a significance probability *p*-value of 0.0000. Thus, the original hypothesis that the coefficient of the correlation of the dependent variable is 0 is rejected, indicating the validity of using the MVP model. According to Table 7, it can be found that compared to the *Z* region, the *U* regions and *V* regions have a weaker preference for Scheme I, and the *U* region has a significantly lower

preference for Scheme I than the Z region. This is due to the fact that in the CLR process, the *U* regions are among the largest beneficiaries of CLR, gaining a higher preference for Scheme I compared to *V* or *Z* and gaining a greater space for the development of construction land; hence, financial compensation is not these regions' main concern.

**Table 7.** Heterogeneity regression results of the types of land use planning.

| Variable | (1) | (2) | (3) |
|---|---|---|---|
| | $Y_1$ | $Y_2$ | $Y_3$ |
| U | −0.3489 ** | 0.2311 * | 0.0432 |
| | (0.1549) | (0.1314) | (0.1310) |
| V | −0.1212 | 0.2773 ** | 0.0320 |
| | (0.1315) | (0.1126) | (0.1113) |
| DO | −0.0011 | −0.0038 * | −0.0026 |
| | (0.0024) | (0.0020) | (0.0020) |
| CS | 0.2036 *** | 0.0338 | 0.0686 * |
| | (0.0426) | (0.0360) | (0.0362) |
| EP | −0.2015 *** | 0.0273 | 0.0637 * |
| | (0.0395) | (0.0348) | (0.0352) |
| LnLCT | −0.1772 *** | 0.2176 *** | 0.1609 *** |
| | (0.0513) | (0.0443) | (0.0435) |
| LnLCV | −0.0276 | −0.0039 | −0.0250 |
| | (0.0357) | (0.0299) | (0.0296) |
| LnPD | 0.1609 | 0.3458 *** | 0.0837 |
| | (0.1152) | (0.0983) | (0.0956) |
| DP | 0.0030 ** | 0.0017 * | 0.0006 |
| | (0.0012) | (0.0009) | (0.0010) |
| GEN | −0.0326 | −0.0542 | −0.1033 * |
| | (0.0663) | (0.0548) | (0.0547) |
| AGE | 0.1423 *** | 0.0622 | 0.0136 |
| | (0.0455) | (0.0382) | (0.0384) |
| EDU | −0.0996 ** | 0.1404 *** | −0.0410 |
| | (0.0454) | (0.0370) | (0.0370) |
| HI | 0.1079 *** | 0.1030 *** | 0.1327 *** |
| | (0.0413) | (0.0349) | (0.0347) |
| HSS | 0.0003 | 0.0010 | 0.0016 * |
| | (0.0011) | (0.0009) | (0.0009) |
| Constant | −0.5304 | −3.9364 *** | −1.4394 * |
| | (0.9096) | (0.7836) | (0.7639) |
| Draws | 50 | | |
| Atrho21 | −0.2189 *** | | |
| | (0.0399) | | |
| Atrho31 | −0.2284 *** | | |
| | (0.0408) | | |
| Atrho32 | 0.1742 *** | | |
| | (0.0343) | | |
| Wald | 253.41 *** | | |
| Likelihood ratio test | 74.8275 *** | | |
| Observations | 2192 | | |

Note: Robust standard errors in parentheses, ***, ** and * indicate significance at the 1%, 5% and 10% levels, respectively.

Regions *U* and *V* both showed significantly stronger preferences for Scheme II than the *Z* regions. This is because the *U* region is more efficient for construction land output and can drive the economic and social development of the region. The *V* region itself is self-reducing and self-using and also prefers to develop the local economy. Region *U* and *V* have a stronger preference for Scheme III than region *Z*, but this is insignificant. Overall, region *Z* has a stronger preference for Scheme I, while regions *U* and *V* are more inclined towards Scheme II.

In terms of economic development realities, regions $U$ and $V$ have a good foundation and comparative advantages in terms of location and efficiency of construction land output and are priority regions for development, while region $Z$ is the focus of CLR, but not the focus of development. As a result, while developing regions $U$ and $V$, it is important to pay attention to the development of region $Z$.

## 6. Conclusions and Policy Implications

### 6.1. Conclusions

CLR in suburban areas is a way to solve the contradiction between the supply and demand of construction land, but it also limits the development of NRACL, which in turn reduces the local residents' support for CLR policies and their choice of compensation schemes. This paper establishes an analytical framework for residents' selection behavior for compensation schemes. By analyzing the task flow, quota flow, financial flow and benefit flow in the CLR process, this paper proposes compensation schemes for the benefits of the direct subjects being reduced in the CLR process and uses the MVP model to study the residents' selection behavior of these compensation schemes. It is found that (1) in the CLR process, CLR tasks are decomposed from top to bottom (municipal government → district government → township government), while land quotas formed by CLR are collected, managed and deployed for use from bottom to top (township government → district government → municipal government), and CLR expenditures are expanded from top to bottom (municipal government → district government → township government) and passed on. (2) In the process of the decomposition of CLR tasks at each level, coordinated use of quotas at each level and transmission of compensation expenditures at each level, the development interests of direct and indirect subjects whose construction land is reduced are affected. (3) In order to protect the development interests of NRACL, there can be three possible solutions: direct economic compensation type, in situ and off-site enhancements of development capacity types. (4) From the macro level, the reasonableness of compensation standards, township population density and development pressure strengthen residents' preference for the direct economic compensation type scheme. Township employment pressure and township location weaken residents' preference for the direct economic compensation type scheme. Township location, population density and development pressure cultivate residents' preference for the in situ enhancement of the development capacity type scheme. The reasonableness of compensation standards, township employment pressure and township location cultivate residents' preference for off-site development capacity enhancement programs. (5) From the perspective of micro factors, residents of higher age, lower education level and higher household income prefer direct financial compensation schemes. Residents with a higher education level and a higher household income prefer in situ development capacity enhancement compensation schemes. Residents with a higher household income and higher family support pressure prefer off-site development capacity enhancement compensation schemes. (6) At this stage, there is no significant difference in the choice of compensation schemes between cadres and non-cadres. (7) Areas with a net reduction in planning prefer direct economic compensation schemes, while areas with a net increase in planning and areas with basic balance in planning prefer in situ development capacity enhancement schemes.

### 6.2. Policy Implications

Based on the findings of the study, the following insights for improving CLR compensation policies are drawn: (1) In the new era and in the process of CLR for urbanization, it is necessary to allow NIACL to continue to develop and NRACL to give play to their advantage of backwardness and avoid the Matthew effect of economic development. In the process of CLR promotion, we should pay attention to the realization of NRACL's development benefits, reserve space for NRACL's development, support village collectives to establish a long-term "blood-making mechanism" and formulate a more systematic policy for CLR benefits. (2) Government departments should fully respect the development

wishes of NRACL in the process of formulating CLR compensation policies and effectively protect the land rights and interests of NRACL and their long-term interests. Specifically, in the process of formulating compensation policies, close attention should be paid to the influence of macro and micro factors on residents' policy choice preferences. (3) At the present stage, residents' demands for compensation are mainly "short-term" in nature, such as direct compensation for economic losses, while they are not sufficiently aware of the need to enhance development capacity in situ and off-site. In order to improve the output efficiency of CLR, the transaction radius of CLR quota allocation needs to be expanded. The reasonableness of compensation standards and the expansion of employment absorption capacity should be enhanced to increase residents' support for off-site development. At the same time, a certain amount of construction land should be reserved for NRACL to be used for industrial development and upgrading and to make use of technology "marriage" in developed regions to drive up the technology level of NRACL. (4) At this stage, there is no significant difference between the development needs of cadres and the residents of NRACL. The economic development of a place has a lot to do with the expectations and efforts of cadres, who show a negative attitude and face difficulties in development. Thus, it is necessary to increase publicity and optimize the cadres' assessment mechanism to enhance the ability to realize the development potential of NRACL. (5) While planning the development of NRACL and basically balanced regions, attention should be paid to planning the development of NRACL.

From the perspective of the realization of development interests in CLR areas, especially the realization of development interests in NRACL, this paper studies the residents' selection behavior of CLR compensation schemes and analyzes the heterogeneous influence of resident status and the types of land use planning, which is enlightening for the improvement of CLR policies. Due to the difficulty of data collection, this paper neither investigates senior-level cadres, such as district-level cadres, nor explores the heterogeneous influence of different levels of cadres as a research direction that can be expanded in the future. Although the sample of this study is from the practice of CLR in Shanghai, China, the findings of this paper can also provide references for other cities that are implementing CLR and will soon implement CLR to enhance the sustainability of CLR policies. In addition, this paper can also provide references for other countries and regions that will soon adopt similar land governance tools to improve their construction land reduction policies.

**Author Contributions:** Conceptualization, J.L., K.W. and H.L.; methodology, J.L., K.W. and H.L.; investigation, J.L., K.W. and H.L.; writing—original draft preparation, J.L., K.W. and H.L.; writing—review and editing, J.L., K.W. and H.L.; funding acquisition, K.W. and H.L. All authors have read and agreed to the published version of the manuscript.

**Funding:** This work was supported by the National Office for Philosophy and Social Science of China, grant number 22AGL027; the Shanghai Planning Office of Philosophy and Social Science, grant number 2020BJB010; the Technology Innovation Center for Land Spatial Eco-restoration in the Metropolitan Area, MNR, Shanghai, 200003, grant number CXZX202201; and the Fundametal Research Funds for the Central Universities, grant number 2022110229.

**Institutional Review Board Statement:** Not applicable.

**Informed Consent Statement:** Not applicable.

**Data Availability Statement:** Not applicable.

**Conflicts of Interest:** The authors declare no conflict of interest.

## Notes

[1]   See http://www.mnr.gov.cn/gk/ghjh/201811/t20181101_2324898.html, accessed on 1 November 2022, for more information.

[2]   See http://www.beijing.gov.cn/gongkai/guihua/wngh/cqgh/201907/t20190701_100008.html, accessed on 1 November 2022, for more information.

[3]   The "198 area" is the existing industrial land outside the planned industrial zone and the planned centralized construction area, covering an area of about 198 square kilo-meters, so named because of the number of areas.

4    See http://jinshan.gov.cn/ghzyj-ghjh/20220105/825960.html, accessed on 1 November 2022, for more information.

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
