# Peer review of "Residents’ Selection Behavior of Compensation Schemes for Construction Land Reduction: Empirical Evidence from Questionnaires in Shanghai, China"

_land, doi:10.3390/land12010020_

Round 1

Reviewer 1 Report

This paper addresses an interesting and important topic; however, writing and logic need further improvement.

 1. Compensation scheme I: please add detailed information about the rules of direct economic compensation, e.g. what is the compensation rate and compensation range under normal circumstances, what is the increased compensation rate, and what does the expanded compensation range include?

 2. Compensation scheme II: The direct subject of the compensation plan is the NRACL residents. What is the specific compensation method for the residents under compensation plan II?

3. Compensation scheme III: why does the author divide the beneficiaries into “the government department at the township level and below (i.e., the townships and underlying subjects)” and “the government department at the district level and above”. After the reduction of construction land, to improve the level of economic development in the region, the reduction indicators will be used first in the construction of the township and similar construction areas, and the remaining indicators will be used for inter-regional balance , that is, when the reduction index is more than the demand for the township construction, the excess index will be purchased by the city or district (county).

4. How to define Compensation Standard in H2 (line 440 on page 13) ? Does the compensation standard here refer to the compensation standard of Compensation Scheme I?

5. H7 is not clear.

6. The maximum value of the variable HHS in Table 3 descriptive statistics is 100, which is unreasonable

7. Compensation Plan III refers to off-site employment, so the factors that affect NRACL residents’ off-site engagement should be included. For example, the working conditions of individual residents or the proportion of non-agricultural household employment, willingness to go out to work, distance factors, etc.

8. Empirical results need to be compared and discussed with previous literature.

9. Heterogeneity can be analyzed by group regression or as an interaction term. For example, divide the samples into two groups according to whether NRACL residents have the status of cadre, and analyze the differences in the influence of various factors on residents' choice behavior in the two groups of samples; or include the interaction term between each factor and the cadre identity variable separately, and analyze the results of the interaction term.

Reviewer 2 Report

The article presents a methodology that is suitable for both MDPI Land and the special issue: “Urban and Rural Land Development and Redevelopment in the Process of Urbanisation”, in fact, it fits perfectly.

English language is good, the article is well organised, explains everything in detail and presents and valid methodology. Results are interesting and can have real world application. Personally I found the article to be very interesting and enlightening.

There are a few small issues that I need to be addressed, most of them being what it feels like a lack of references. For most of them, I’m looking to either see references being added or justifications why there is no need for a reference.

First and foremost is import to define the concept of Construction Land Reduction. A well presented definition should appear in the Introduction’s first paragraph, with adequate references, such as: https://www.sciencedirect.com/science/article/abs/pii/S0264837721000338
https://onlinelibrary.wiley.com/doi/abs/10.1111/grow.12532 and more!!
“Construction land reduction is a land restoration tool that reclaims inefficient, dispersed, and heavily polluted construction land outside urban concentrated construction (…)”. All of this part should be in the main text and not as a footnote, again not forgetting the correct references to all of it.

LINE 29: “The suburban areas in economically developed regions of China are still in a stage of rapid urbanization and development, and the contradiction between supply and demand of construction land remains very prominent.” - Needs references, both for the rapid urbanisation and development and for the contradiction between supply and demand. Doesn’t has to be from other articles but at least from official government authorities that in a direct or indirect way confirms it.

LINE 31: “China implements a system of spatial use planning and land use control, and controls the total amount and intensity of construction land, and the control of construction land constrains further economic development.” - Again, it needs references and examples from municipal plans and local government authorities that confirm this.

LINE 35: “In order to solve the contradiction between supply and demand of construction land, it can be achieved” - Needs a small rephrasing to improve phrase connection, as in order to doesn’t really match the “it can be (…)” part.

LINE 38: “In urban areas of economically developed regions, it is mainly the former; while in suburban areas, it is mainly the latter.” - Again, references for both the former and latter.

LINE 47: Urban Master Plan of Beijing, China (2016-2035) this needs to be referenced, even if its can’t be accessed by everyone.

Line 50: “While CLR solves the contradiction between supply and demand of construction land, it also causes a slowdown in the development of net reduction areas of construction land (NRACL) and a change in the interest demands of residents in CLR areas.” - Again, needs references!

LINE 79: “In the early stage of the implementation of CLR policy, the direct loss of interests was used as the basis of compensation, and the objects of compensation were the direct damaged subjects, without considering the indirect losses and indirect damaged subjects; the basis of being compensated and the objects of compensation were not comprehensive. With the expansion of the scale of the CLR and the extension of the period, its indirect impacts are becoming more and more obvious, especially the loss of local employment opportunities is obvious.” - At this point I’m not sure if this is your opinion/analysis of the implementation of the CLR policy or actual conclusions taken that needs, once again, to be referenced.
The remaining paragraph follows the same thing. Is this the authors analysis? Is this documented? The word “obvious” is used twice in the same paragraph. This is not the most correct phrasing as what can be obvious to some, might not be scientifically obvious. Again, is it obvious as the authors analysis or it can be referenced as something obvious?

LINE 111: “Urbanization is also a process of expansion of construction land and concentration of population to cities” - Needs references.

LINE 122-146: I would recommend bullet points on different lines for better understanding. Visually separating the bullet points significantly helps readability.

LINE 141: “Land reconfiguration is a spatial planning process (…)” - Although land reconfiguration is not a complex concept/notion it would be interesting to add what it means (with adequate references)

LINE 149: “The purpose of CLR is to develop and improve people's well-being. But the current operation of CLR has brought about the problem of unequal benefits of development dividends, especially unequal opportunities for development and damage to the interests of residents in NRACL.” - This can be only one sentence.

LINE 159: “The established literature has less research on CLR, (…)” - Rewrite in a better way. Something like: There is gap in current/existing literature when it comes to CLR” or similar.

Footnote 2: I believe it is important enough not to be a footnote but be on the main text.

LINE 176: “The CLR process involves three levels of government - municipal, district and town-ship - and three types of underlying subjects. Village collectives, land enterprises and residents are the base subjects most affected by CLR.” - Need references

LINE 247: Again, for improved readability I would suggest bullet points on different lines.

LINE 308: “In the process of CLR, how can the development interests of the reduced direct subjects be realized? Based on theoretical analysis and combined with CLR practice in Shanghai, China, this paper summarizes three compensation schemes for safeguarding the development interests of NRACL.” - I would suggest rewriting without the question-mark.

Table 1: It would be ideal to add the first line (title line) to the second page of the table. Since it’s a big table and uses two pages, it would improve readability to also have the header on the second page.

LINE 519: Perhaps creating a table with all the hypotheses to improve readability.

LINE 572: “Since the use of construction land is also affected by land use planning in the CLR process, for this reason, the influence of land use planning type is considered in this paper.” - Needs minor english corrections

LINE 576: “construction land area9: the first type” - Typo of a 9 not belonging there.

LINE 759: “(11) Households with higher household incomes have higher expectations of economic development and thus have a higher preference for schemes I, II and III.” - This means that higher household incomes prefer all of the three schemes available? What do you mean by that? What I’m understanding is that they prefer all the three available schemes? Please clarify this.

LINE 784: “this are10: At this stage, the performance” - Similarly to line 576 it appears a number out of nowhere.

Footnote 10: I would say that this is important enough information to be on the main text and not as a footnote. If I’m understanding correctly this is a small methodology limitation.

6. Conclusions and Policy Implications - What has been written is good but I would recommend to sum it all up on one last paragraph. The article ends too abruptly. Additionally, what were the methodology and research limitations? And what further work can be done in the field, testing new policies, new schemes or new scenarios? I think summing all up at the end, as well as answering these questions would largely improve the last chapter.

Round 2

Reviewer 2 Report

All my comments were address and all the necessary changes were made. As I previously mentioned the article presents a methodology that is suitable for both MDPI Land and the special issue: “Urban and Rural Land Development and Redevelopment in the Process of Urbanisation”, in fact, it fits perfectly.